# Insight into Prevention of Neisseria Gonorrhoeae: A Short Review

**DOI:** 10.3390/vaccines10111949

**Published:** 2022-11-18

**Authors:** Vincenza La Fauci, Daniela Lo Giudice, Raffaele Squeri, Cristina Genovese

**Affiliations:** Department of Biomedical and Dental Sciences and Morphofunctional Imaging, University of Messina, 98124 Messina, Italy

**Keywords:** *Neisseria gonorrhoeae* (gonococcus), *Neisseria meningitidis*, gonococcal infection, vaccines, meningococcal, 4CMenB, VA-MENGOC-BC, MenBvac, MeNZB

## Abstract

*Neisseria gonorrhoeae* (gonococcus) and *Neisseria meningitidis* (meningococcus) are important global pathogens which cause the sexually transmitted diseases gonorrhea and meningitis, respectively, as well as sepsis. We prepared a review according to the preferred reporting items for systematic reviews and meta-analyses (PRISMA), with the aims of (a) evaluating the data on the MenB vaccination as protection against sexually transmitted infections by *N. gonorrhoeae* and (b) to briefly comment on the data of ongoing studies of new vaccines. We evaluated existing evidence on the effect of 4CMenB, a multi-component vaccine, on invasive diseases caused by different meningococcal serogroups and on gonorrhea. Non-B meningococcal serogroups showed that the 4CMenB vaccine could potentially offer some level of protection against non-B meningococcal serogroups and *N. gonorrhoeae*. The assessment of the potential protection conferred by 4CMenB is further challenged by the fact that further studies are still needed to fully understand natural immune responses against gonococcal infections. A further limitation could be the potential differences between the protection mechanisms against *N. gonorrhoeae*, which causes local infections, and the protection mechanisms against *N. meningitidis*, which causes systemic infections.

## 1. Introduction

*Neisseria gonorrhoeae* (gonococcus) and *Neisseria meningitidis* (meningococcus) are important global pathogens causing gonorrhea and meningitis, respectively, and/or sepsis [1].

*Neisseria meningitidis* leads to an invasive meningococcal disease (IMD) that could be fatal and lead to severe morbidity, and it is widespread all over the world. It can be characterized into 12 serogroups, of which the most common are serogroups A, B, C, W135, X, and Y [2]. Serogroup B (MenB) has been predominant in children and young adults, and it is broadly diffused, especially in Europe. Luckily, recently protective vaccines against B serogroup have become available. In fact, according to ECDC data in Italy for *Neisseria gonorrhoeae* (gonococcus) reported in 2019, there were 813 confirmed cases, with a higher prevalence in the age groups of 15–24 and 25–34 years [3]. Meanwhile, in Italy in 2018, according to the College of Health, the number of new cases was 601 [4]. Due the importance of this pathogen worldwide, two protein-based vaccines, namely 4CMenB (Bexsero^®^) and rLP2086 (Trumenba^®^), are currently licensed for use in different countries. In other countries, the VA-MENGOC-BC vaccine is available [5]. Given this premise, it is extremely important to choose the right vaccine for the prevention of this condition; furthermore, considering the epidemiological data concerning this pathogen in Europe, it is essential to choose which vaccine is best for a booster dose.

*N. gonorrhoeae* is a significant health problem worldwide due to the rapid change in antimicrobial susceptibility with the development of antibiotic resistance to the molecules generally used for its treatment, the presence of limited studies, and the absence, to date, of an effective vaccine [6].

According to data from the World Health Organization (WHO), the incidence rates translate globally into 86.9 million (95% UI: 58.6–123.4 million) new gonorrhea cases in women and men aged 15–49 years [7]. From an epidemiological point of view, gonorrhea appears to be the second-most sexually transmitted disease, with approximately 1.6 million new gonococcal infections in the United States in 2018, especially in young adults [8].

Although it occurs asymptomatically in 10–50% of cases, antimicrobial resistance (AMR) in many countries has been extremely high in recent years with a lack of efficacy, not only in the current recommended treatment for gonorrhea (ceftriaxone and azithromycin), but also in penicillin, sulfonamides, tetracyclines, fluoroquinolones, and macrolides. Gonococcal infections have critical implications for reproductive, maternal, and neonatal health, including a five-fold increase in HIV transmission, infertility, and inflammation, leading to acute and chronic lower abdominal pain in women, ectopic pregnancies and maternal death, first-trimester abortions, and severe neonatal eye infections that can lead to blindness.

The financial costs of these complications are very high for both individuals and healthcare systems. Antimicrobial resistance increases this burden by prolonging the infection in more people and increasing the number of people with long-term complications. To date, prevention involves the use of prophylaxis and the avoidance of direct sexual contact [9], given the absence of an effective vaccine; a recent review of the literature illustrates the main scientific efforts in this regard [9]. At present, only two gonococcal vaccine candidates have reached the stage of human experimentation in the past. The lack of a clear correlation between protection and the absence of immunity to Ng reinfection have called into question the possibility of a gonorrhea vaccine. However, several studies have been performed in order to find the right antigens for the development of an effective vaccine.

Considering all the problems reported today, it is important to: (a) explore the possibility of avoiding sexually transmitted diseases through available vaccines; and (b) improve randomized clinical trials on new and existing vaccines. From this point of view, several vaccines are currently available for the prevention of Neisseria meningitis B, which have shown cross-protection for N. gonorrhoeae and which could be used until an effective vaccine is discovered. In particular, the vaccines available today are based on outer-membrane vesicles or proteins. For the first type, we recall the following three vaccines: VA-MENGOC-BC [Finlay Institute]; MenBvac [Norwegian Institute of Public Health]; MeNZB [Novartis], which are used in Cuba, Norway, and New Zealand, respectively, and are specifically designed for specific strains during outbreaks, resulting in poor coverage against other MenB strains. The second ones are given by MenB-4C (Bexsero^®^) and MenB-FHbp (Trumenba^®^).

The first vaccine was licensed in the USA in 2015 and is available for people aged 12–25 years. It is a four-component vaccine resulting from the reverse transcription of the complete genomic sequence of the pathogen strain MC58, containing: (1) the fHbp 1.1 variant (subfamily B) fused to the genome-derived *Neisseria* (GNA) 2091 antigen; (2) adhesin *Neisseria* A (NadA); and (3) neisserial heparin-binding antigen (NHBA) peptide 2 fused with GNA1030, and this results in increased bactericidal activity. The recommended dosing schedule for 4CMenB in humans aged >2 years is two 500 μL doses given intramuscularly, one month apart [10,11].

The 4CMenB vaccine offers broad protection against invasive MenB by inducing each of the bactericidal antibody antigens that mediate the killing of strains, also acting cooperatively [12,13,14].

Furthermore, antibodies induced by immunization with 4CMenB can recognize different epitopes on each of the fHbp, NadA, and NHBA antigens, further amplifying bactericidal activity through this synergism [15].

The fHbp2086 vaccine consists of equal amounts of two factor-H-binding protein (fHbp) variants belonging to subfamilies A and B, which were identified by biochemical approaches [16,17].

As such, the aim of the present research is to describe the available data on the MenB vaccination as protection against the STI caused by *N. gonorrhoeae*.

## 2. Materials and Methods

We prepared a review according to preferred reporting items for systematic reviews and meta-analyses (PRISMA).

We searched the main scientific libraries (PubMed, including MEDLINE, Web of Science, and Embase) for all studies reporting a reduction in *N. gonorrhoeae* infection in articles indexed up to the date of the search, with no language restrictions. Table 1 shows the keyword search strategy for one database, Web of Science. Studies were included if they investigated vaccine-induced immunity in healthy individuals who received a MenB vaccine, including different dosages and the time points of vaccine administration. We excluded all case studies/reports, letters to editors, review papers, personal opinions, and any other type of study with inconsistent data or which did not report original data. We also conducted hand searches of the reference lists of the included studies and related reviews. We exported all studies retrieved from the electronic searches for deduplication and screening. Two review authors (CG and RS) independently screened the titles and abstracts to identify potentially eligible studies, and any disagreements between the two authors were resolved by discussion and consensus. We obtained the full texts of all potentially eligible studies. Two authors (VLF and CG) independently screened the full texts and identified the included studies, resolving discrepancies through discussion and consensus.

### Data Extraction

Four independent authors (CG, VLF, DLG, and RS) identified potentially relevant articles and collected the following data: first author’s last name; year of publication; study design; total number of participants; age range if applicable; and human or mice study. 

Of the 120 records identified, after removing duplicate records, we selected 106 for a full text review and, finally, 20 were discussed.

## 3. Results

We will discuss the available data on the prevention of gonorrhea by MenB vaccines. The included studies were reported in Table 2. 

### 3.1. Available Data on the Prevention of Gonorrhea by MenB Vaccines

#### 3.1.1. Immunogenicity of Gonococcal OMVs and the 4CMenB Vaccine against Gonorrhea in Mice

*Neisseria gonorrhoroeae* is not an animal pathogen, but a human one, and its investigation is obtained through the use of a murine model given by the infection of the genital tracts of mice treated with estradiol. Moreover, its action, similarly to *N. meningitidis*, depends on the K antigen and on endotoxins (outer-membrane lipopolysaccharide). Due to these reasons, under-cited studies were performed on these premises.

Leduc et al. [18] tested the hypothesis that the external-membrane-vesicle vaccine 4CMenBV (Bexsero) against *Neisseria meningitidis* induces cross-protection against gonorrhea in a female mouse model infected in the genital tract. They found that immunization with half the dose of the vaccine significantly accelerated clearance and reduced the bacterial load of Ng compared to the administration of a placebo. In addition, an increase in serum IgG, and in vaginal IgA and IgG, was also obtained both in the administration through the subcutaneous (SC) and intraperitoneal (IP) routes, leading to clearance within one week in the vaccinated group compared to the control group (injected with alum or PBS). This finding directly supports the evidence available today relating to cross-protection induced by vaccination with the production of protective antibodies against various surface antigens. Furthermore, they found that 4CMenB reproducibly accelerates the clearance of Ng from the murine genital tract and reduces bacterial load over time, and that complement-mediated and opsonophagocytic bacteriolysis may contribute to protection [18]. In particular, complements are the main modality used by our immune system to counteract infection.

In the study of Plante et al., the intranasal administration of lithium-chloride-extracted gonococcal outer-membrane preparations to female mice (treated with estradiol to prolong infection) was performed to evaluate the impact on vaginal colonization [19]. The results were an accelerated clearance of *N. gonorrhoeae* compared to the control group and the detection of gonococcal-specific antibodies in the sera of the immunized mice [19].

The same results were also reported in a study investigating the intravaginal administration of gonococcal OMVs with microencapsulated interleukin-12 in female mice [20].

In summary, the demonstration that an authorized NMV OMV-based vaccine accelerates the clearance of Ng in a mouse model’s genital tract infection is direct evidence that cross-species protection can be an effective vaccine strategy for gonorrhea.

#### 3.1.2. Ecological Studies

Several ecological studies have shown the protection of people immunized with OMV MenB for *N. gonorrhoeae*. In Norway and Cuba, a reduction in gonorrhea cases was identified after MenB OMV vaccines [20,21]. In Norway, the rates of gonorrhea had already reduced since the mid-1970s, during which there was a clear decline in both sexes within subjects aged 20 to 24 years. A limit of the study is the presence of behavioral factors, such as the use of condoms since the early 1990s, especially among young people [20].

In Cuba, the authors both reviewed epidemiological data for *N. meningitidis* and *N. gonorrhoeae* infections and collected serum, saliva, and oropharynx samples from high school students who had previously been vaccinated with VA-MENGOC-BC (MBV) during their infancy. They were revaccinated with a third dose during the study. The epidemiological data showed a decrease in the incidence of gonorrhea. The samples, tested by Western blot analysis, were positive with an increase in the serum responses of the anti-MBV (PL) antigens, except for the anti-PL IgA responses of saliva present only and significantly induced in the carriers. Carriers were augmented with a third dose of MBV-induced, similar anti-gonococcal responses and serum-PL saliva IgA and IgG [21]. Further epidemiological data show the influence of MBV on the incidence of gonorrhea, suggesting that the vaccines depend on the age of sexual arrival [22].

In Norway, a retrospective case-control study was conducted on patients in sexual health clinics aged 15–30 who were eligible to receive MeNZB. Cases were identified through the laboratory isolation or detection of *Neisseria gonorrhoeae* from a clinical specimen, while the controls were individuals with a positive chlamydia test. The estimated vaccine efficacy after adjustment for ethnicity, deprivation, geographic area, and gender was 31% (95% CI 21–39) [23].

A retrospective cohort study was conducted on a cohort of individuals born between 1984 and 1999 (*n* = 935,496) to estimate the efficacy of the New Zealand meningococcal B vaccine against hospitalization associated with gonorrhea. The efficacy of the vaccine (MeNZB™) against hospitalization caused by gonorrhea was estimated to be 24% (95% CI 1–42%), suggesting a significant reduction in the hospitalization rate for gonorrhea. This supports previous research, indicating a possible cross-protection of this vaccine against gonorrhea acquisition and disease in an outpatient setting [24].

#### 3.1.3. Ongoing Studies Assessing Impact of 4CMenB against *N. Gonorrhoeae* and Real-World Studies

A phase 3, double-blinded, randomized placebo-controlled, multi-centered trial is being conducted in Australia to assess the efficacy of the 4CMenB (Bexsero^®^) vaccine, in the prevention of *Neisseria gonorrhoeae* infection. The sample comprises people aged 18–40 years old, men, transgender women, and gay or bisexual men+, who are either HIV negative and taking pre-exposure prophylaxis [PrEP], or HIV positive with a low viral load (<200 copies/mL) and CD4 count >350 cells/cmm, who have a high *N. gonorrhoeae* incidence and are indicated by Australian guidelines to have regular, comprehensive sexual health screenings [25].

Another phase II multi-center study with a length of 24 months is currently being conducted in the USA and Thailand to verify the efficacy of two doses of 4CMenB in the prevention of urogenital and anorectal infections in at-risk subjects aged 18 to 50 years (*n* = 2200) [26].

In the USA, a phase 2 case-control clinical trial is being conducted to assess the systemic and mucosal immunogenicity of the 4CMen vaccine against *Neisseria gonorrhoeae*, using a placebo vaccine for comparison (arm 1 = 40 e arm 2 = 10; *n* = 50). The enrolment will be stratified by both sex and treatment arm, and the length of the study is estimated to be 14 months. The endpoint is to detect the rectal mucosal Immunoglobulin G (IgG) antibody response to *Neisseria gonorrhoeae* elicited by the 4CMenB vaccine compared with the placebo vaccine [27].

Another clinical trial to demonstrate that the meningococcal B vaccine (Bexsero^®^) reduces the occurrence of the first episode of *Neisseria gonorrhoeae* is being made in France. Other objectives include a reduction in the occurrence of cumulative episodes of NG, first episode of anal and urinary NG, first episode of symptomatic anal and urinary NG; serum bactericidal activity against meningococcus B and gonococci; and the tolerance; prevalence, and incidence of meningococcal carrying at the pharyngeal, anal, and urinary levels, and the impact of prophylaxis on it [28].

In Australia, a study aims to evidence the impact and the effectiveness of the 4CMenB vaccine against IMD and gonorrhea in people aged 14–19 years with a target sample of 7100 participants, and the estimated completion date is 31 December 2024 [29]. One of the objectives is to implement 4CMenB immunization in young people aged 14–19 years, a second one is to evidence carriage prevalence; and finally, one is to study the vaccine effect (impact and effectiveness) against both invasive meningococcal disease (IMD) and gonorrhea using data from the above study, comparing notifications between vaccinated and unvaccinated participants [29].

Wang et al. [30] are carrying out a cohort and case-control study among patients adhering to the 4CMenB vaccination program in South Australia, obtaining data on both disease notifications and vaccination coverage. The efficacy of the vaccine was estimated by the reduction in the probability of infection using screening and case-control methods. The impact of the vaccine was estimated using incidence ratios (IRRs), obtained by comparing the number of cases in each year following the start of the vaccination program with cases in the age-equivalent cohort during the years of the pre-vaccination program. Two years after the implementation of the childhood vaccination program, the incidence of serum meningococcal disease of group B was reduced in the implementation of the program in children aged 1 to 11 months, but not the remaining ages. Meanwhile, the two-dose vaccine efficacy was estimated to be between 94.2% and 94.7% for invasive meningococcal disease B and 32% for gonorrhea [31].

Another study is being conducted in South Australia in adolescents and young adults 15–24 years of age to evaluate the effectiveness of the 4CMenB vaccination program against invasive meningococcal disease and gonorrhea through a combination of observational studies using routine surveillance and research data [31].

In a study conducted in Canada, after the vaccination of 82% of subjects between the ages of 2 months and 20 years, no cases of gonorrhea occurred among the vaccinated, while two cases were reported among those not vaccinated, obtaining a significant effect in multivariate analysis (relative risk of B-IMD: 0.22; *p* = 0.04) [32].

Similar values, with lower vaccination coverage, were reported in a US case-control study reporting a vaccine efficacy in protection against gonorrhea of 40% with two doses and of 26% with one dose [33].

In a matched-cohort study conducted from 2016 to 2020 [34] at Kaiser Permanente Southern California, the authors examined the association of gonorrhea and chlamydial infection between 6641 recipients of 4CMenB and 26,471 recipients of MenACWY in which their findings, during the follow-up period, showed a reduction in the incidence rates of gonorrhea by 46%, while no effect was found for Chlamydia.

A bioinformatic analysis was performed to assess the similarity of MeNZB OMV and Bexsero antigens to gonococcal proteins, and it was demonstrated that Bexsero induces antibodies in humans that recognize gonococcal proteins that may provide additional cross-protection against gonorrhea [35].

## 4. Discussion

Our study showed the advantages of vaccination for *Neisseria meningitidis* B in the prevention of gonorrhea infection, and this is important considering the reduction in the burden of the disease and the dependence on antibiotic treatment. We must remember that MenZB is no longer on market, but the antigenic content of OMV is equivalent to that of 4CMenB [35]. Va-MenB is at this time only used in Cuba [21].

Younger people aged 10–24 years and/or specific populations have an increased risk of gonococcal infection, and they are the main target for this vaccine, although a favorable safety profile will need to be demonstrated in adults before moving on to adolescents. 

Moreover, *N. gonorrhoeae* and *N. meningitidis* are genetically similar with the presence of NHBA and OMV in both pathogens. For NHBA, they have a similarity of 67% and are highly preserved in *N. gonorrhoeae* (identity > 93%). Additionally, for OMV, there is a similarity of more than 90% for FbpA, FrpB, Tbp1, OMP85 and OMP P1, and for MtrE and NspA. Furthermore, *Neisseria gonorrhoeae* has a similarity of more than 80% for LptD and proteins with a binding domain of LysM with *N. meningitidis* [36,37].

The 4CMenB vaccine components are present and stored in different strains of *N. meningitidis* and *N. gonorrhoeae*. All studies, except the case-control ones, have confirmed the role of this vaccine in the partial protection against gonorrhea.

The duration of protection was not clinically assessed; however, a US study conducted mathematical modelling and sensitivity analyses to assess vaccine effectiveness assuming a 6- and 12-month duration of protection. The study observed that assuming a 6-month or a 12-month duration of protection, complete (two-dose) and partial (one-dose) vaccination series were both significantly protective against gonorrhea with a *p* < 0.001 [33].

In the studies discussed in this review, the immunity studied was exclusively the antibody one, without evaluating the possible responses of the cellular immune system; in fact, the same humoral immunity depends on different aspects in which the potential level of protection offered can vary according to the subject or strain studied.

Further research is needed on the prevention of gonorrhea by antibodies induced by 4CMenB. It should be remembered that the NHBA recombinant fusion protein of *Neisseria meningitidis* group B is present in only one of the vaccines discussed here. This factor plays a fundamental role as it binds heparin, and it seems that it also plays a role in the evasion of the complement system [33,34,35,36,37,38,39,40,41,42], which is the main way used by our immune response to fight *N. gonorrhoeae*. As indicated by other authors, genomic and proteomic characterizations of IMD and gonorrhea isolates are necessary; this could be important for a variety of reasons, such as to provide an insight into the molecular basis of the broad coverage of underlying strains, define the formulation of future non-capsular meningococcal vaccines, and support decisions regarding prevention and immunization programs [17,33,34,35,36,37,38,39,40,41,42].

Furthermore, the results of the studies evaluating the potential protection conferred by 4CMenB against gonorrhoeae must be interpreted with caution because they are based on observational and ecological studies. Furthermore, more studies are still needed to fully understand natural immune responses against gonococcal and meningococcal infections [17,33,34,35,36,37,38,39,40,41].

Gonorrhea is a global public health challenge with an estimated annual 82 million cases among adults in 2020. In 2017–2018, susceptibility decreased and resistance to ceftriaxone was reported by only 31% of countries and resistance to cefixime was reported by 47%. Resistance to azithromycin was reported by 51 (84%) of 61 reporting countries, and ciprofloxacin was reported by all 70 (100%) reporting countries. The annual proportion of decreased susceptibility or resistance across countries was 0–21% to ceftriaxone and 0–22% to cefixime, and that of resistance was 0–60% to azithromycin and 0–100% to ciprofloxacin. The number of countries reporting gonococcal AMR and resistant isolates, and the number of examined isolates, have increased since 2015–2016 [39].

Current attempts to obtain a vaccination against gonorrhea are limited by a lack of correlated protection and the extreme antigenic variability of *Neisseria*. In the past, only two vaccines have gained approval for clinical trials, of which one was based on pilin and the other one was based on whole gonococcal cells killed by heat and partially lysed [40].

For a more comprehensive review of new gonorrhea vaccine developments, the reader is directed towards a recent review conducted by Maurakis et al. [9].

Furthermore, there is a need for randomized controlled trials and additional studies to assess efficacy of OMV-containing MenB vaccines in reducing gonorrhea infections, transmission, and morbidity-associated costs. In the presence of vaccine refusal or vaccine hesitancy, the association with a sexually transmitted infection can affect the acceptability of an anti-gonococcal vaccine; from this point of view, communication must take place in an appropriate manner for the involvement of patients in their own health-related choices.

Finally, the limitations of this review include the type of studies included and their limitations, the absence of study on the persistence of immunity, and the lack of a correlation with protection.

## 5. Conclusions

In conclusion, this review adds information on the pre-existing evidence on the effect of 4CMenB on invasive meningococcal disease and on gonorrhea. Several different studies have tried to predict the coverage of the 4CMenB strain on other infections and particularly on gonorrhea.

## Figures and Tables

**Table 1 vaccines-10-01949-t001:** Research strategy.

TS = ((Vaccines; Meningococcal; Meningococcal Vaccine; Vaccine; Meningococcal), (Neisseria meningitidis; Serogroup B), (Gonococcus neisseri; Micrococcus gonorrhoeae; Merismopedia gonorrhoeae; Micrococcus der gonorrhea; Micrococcus gonococcus; Diplococcus gonorrhoeae, Gonococcus), and (Case-Control Study or Studies; Case-Control or Study; Case-Control or Case-Comparison Studies or Case Comparison Studies or Case-Comparison Study or Studies; Case-Comparison or Study, Case-Comparison or Case-Compeer Studies or Studies; Case-Compeer or Case-Referrent Studies or Case Referrent Studies or Case-Referrent Study or Studies; Case-Referrent or Study, Case-Referrent or Case-Referent Studies or Case Referent Studies or Case-Referent Study or Studies; Case-Referent or Study; Case-Referent or Case-Base Studies or Case Base Studies or Studies; Case-Base or Case Control Studies or Case Control Study or Studies; Case Control or Study, Case Control or Nested Case-Control Studies or Case-Control Studies; Nested or Case-Control Study; Nested or Nested Case Control Studies or Nested Case-Control Study or Studies; Nested Case-Control or Study; Nested Case-Control or Matched Case-Control Studies or Case-Control Studies; Matched or Case-Control Study; Matched or Matched Case Control Studies or Matched Case-Control Study or Studies; Matched Case-Control or Study; Matched Case-Control or Cohort Study or Studies; Cohort or Study; Cohort or Concurrent Studies or Studies; Concurrent or Concurrent Study or Study; Concurrent or Closed Cohort Studies or Cohort Studies; Closed or Closed Cohort Study or Cohort Study; Closed or Study; Closed Cohort or Studies; Closed Cohort or Analysis; Cohort or Cohort Analysis or Analyses; Cohort or Cohort Analyses or Historical Cohort Studies or Cohort Study; Historical or Historical Cohort Study or Study; Historical Cohort or Cohort Studies; Historical or Studies; Historical Cohort or Incidence Studies or Incidence Study or Studies; Incidence or Study; Incidence or Vaccinations or Immunization; Active or Active Immunization or Active Immunizations or Immunizations; Active or Intervention Study or randomized controlled trial or Clinical Trial, Phase 4).

In accordance with guidelines, no age restriction was used. In addition, to avoid limiting the search results, we did not use the word “humans” to screen references.

**Table 2 vaccines-10-01949-t002:** Synthesis of the included study.

Number of References, Name of the First Author, Year	Target
[18] Leduc, 2020	Mice
[19] Plante, 2000	Mice
[20] Liu, 2017	Mice
[21] Whelan, 2016	Humans
[22] Reyes Diaz, 2021	Humans
[23] Petous-Harris, 2017	Humans
[24] Paynter, 2019	Humans
[25] Clinical trials	Humans
[26] Clinical trials	Humans
[27] Clinical trials	Humans
[28] Clinical trials	Humans
[29] Clinical trials	Humans
[30] Wang, 2022	Humans
[31] Marshall, 2022	Humans
[32] De Wals, 2017	Humans
[33] Abara, 2022	Humans
[34] Bruxvoort, 2022	Humans
[35] Semchenko, 2011	Humans

## Data Availability

No new data were created or analyzed in this study. Data sharing is not applicable to this article.

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
