# Peer review of "Insight into Prevention of Neisseria Gonorrhoeae: A Short Review"

_vaccines, 2022, doi:10.3390/vaccines10111949_

Round 1

Reviewer 1 Report

This article is interesting and topical because it makes a tour of all the publications or clinical trials in progress involving the use of the vaccine targeting meningococcal to treat gonococcal STIs. The literature search is well conducted.

It would have been interesting to retrieve a summary table of the analysis of the 20 publications discussed.

Minor comments:

In all the manuscript: N. gonorrhoeae and N. menigitidis in italic

Line 14: add space before DLG

Line 35: insert comma before and after “in 2019”

Line 36: insert comma before and after “in 2018”

Line 46: In 2016 the global prevalence of this disease as of 30.6 million cases worldwide ac-46 cording to data by World Health Organization.

I suggest that the authors, instead of the prevalence, give the incidence rates translate globally into 86.9 million (95% UI: 58.6–123.4 million) new gonorrhoea cases in women and men aged 15–49 years.

Line 94: corrected N. gonorraeae to N. gonorrhoeae

Line 95: Table 1 was not available

Line 190: deleted “A”

Author Response

Dear Reviewer thanks for the opportunity to make the correction in our study.

We modified the manuscript according to your indications and the indications of the second reviewer. 

Best regards

Reviewer 2 Report

First et all, the work fits more as a short communication than a review.      

Below are examples of more detailed comments.

·        Why do the authors refer to Table 1, but do not include it?

·        What does it mean, “Four independent reviewers ….) (line 140), after all, they are authors and not reviewers.

·        Not everywhere in the text full reference data are given, e.g. line 163.

·        What means “Evidence from the Word”? (line 252, point 3.4). Why it was separated from country?

·        What is the origin of the abbreviation MenB line 69. It means Neisseria meningitidis serogroup B?

In addition, the work needs a thorough editorial correction; eg. names of bacterial species should be written in italics, the literature list is written in a different fonts.

Author Response

Why do the authors refer to Table 1, but do not include it?

-The table is now framed.

What does it mean, “Four independent reviewers ….) (line 140), after all, they are authors and not reviewers.

-We corrected it

Not everywhere in the text full reference data are given, e.g. line 163.

-We add references number 18.

What means “Evidence from the Word”? (line 252, point 3.4). Why it was separated from country?

-Because we created a section for all type of study.

What is the origin of the abbreviation MenB line 69. It means Neisseria meningitidis serogroup B?

-Yes, we correct it.

Round 2

Reviewer 2 Report

Despite the important healthy issue addressed, the manuscript needs a major redesign and completion.

The manuscript results include 3 chapters, one of which should be combined with the other. This would only be an allegation but not the cause of disqualification if the content had been comprehensive and clearly presented. For example, chapter 3.1 is based on 3 original papers, and  3.2 on 4 articles.

The manuscript contains loosely presented information without attempting to connect or interpret it.

There needs to be more information about other searches and attempts to find a vaccine against Neisseria gonorrhoeae, for example, based on phages. The references are sparse, and it includes review papers generally characterizing Neisseria.

Author Response

We thank you for the opportunity to make the revisions requested by reviewer 2 for further improve our submission and we are glad that reviewer 1 appreciated our first round of review.

Below you will find a point-by-point response to the comments raised by the reviewers.

Finally, the manuscript with changes marked will be attached to the MDPI submission system.

We hope that this improved version of the manuscript will be suitable for publication in Vaccines.

The manuscript results include 3 chapters, one of which should be combined with the other. This would only be an allegation but not the cause of disqualification if the content had been comprehensive and clearly presented. For example, chapter 3.1 is based on 3 original papers, and  3.2 on 4 articles. 

We modify it

The manuscript contains loosely presented information without attempting to connect or interpret it. 

We improve it.

There needs to be more information about other searches and attempts to find a vaccine against Neisseria gonorrhoeae, for example, based on phages.

We included other studies

The references are sparse, and it includes review papers generally characterizing Neisseria.

We included other studies

Round 3

Reviewer 2 Report

The manuscript has been improved. However, still, some corrections are needed. 

The main correction, which should be performed: is the title. The title is not consistent with the manuscript content. It should be changed. 

If line 200 is actual (If such was the Authors' intention that this is what the manuscript is about), the title should be changed.  

Why have the Authors limited the manuscript to data from a few countries and not from around the world? Again, this is the issue with the title. 

A new chapter should also be added, including perspectives on obtaining other vaccines against Neisseria gonorrhoeae

Editorial errors. 

  • Bacterial species names should be italicized. 
  • Different type of fonts is in the manuscript.
  • Lines 87, 102, 112, etc. are written in italics, which they shouldn't be.
  • Errors in bacterial species names, e.g., line 106.

Author Response

We thank you for the opportunity to make the revisions requested by reviewer 2 for further improve our submission.

Below you will find a point-by-point response to the comments raised by the reviewer.

Finally, the manuscript with changes marked will be attached to the MDPI submission system.

We hope that this improved version of the manuscript will be suitable for publication in Vaccines.

  • The main correction, which should be performed: is the title. The title is not consistent with the manuscript content. It should be changed. If line 200 is actual (If such was the Authors' intention that this is what the manuscript is about), the title should be changed.  Why have the Authors limited the manuscript to data from a few countries and not from around the world? Again, this is the issue with the title . We change the title

  • A new chapter should also be added, including perspectives on obtaining other vaccines against Neisseria gonorrhoeaeIt is not the aim of our study. However we add some comments in the discussion section.

Editorial errors. 

  • Bacterial species names should be italicized. We corrected it
  • Different type of fonts is in the manuscript. We corrected it
  • Lines 87, 102, 112, etc. are written in italics, which they shouldn't be. We corrected it
  • Errors in bacterial species names, e.g., line 106. We corrected it